# Kidney function on admission predicts in-hospital mortality in COVID-19

**Sinan Trabulus[1], Cebrail Karaca[1], Ilker Inanc Balkan[2], Mevlut Tamer Dincer[1], Ahmet Murt[1], Seyda Gul Ozcan[3], Rıdvan Karaali[2], Bilgul Mete[2], Alev Bakir[4], Mert Ahmet Kuskucu[5], Mehmet Riza Altiparmak[1], Fehmi Tabak[2], Nurhan Seyahi[1]\***

**1** Department of Nephrology, Cerrahpasa Medical Faculty, Istanbul University – Cerrahpasa, Istanbul, Turkey, **2** Department of Infectious Diseases and Clinical Microbiology, Cerrahpasa Medical Faculty, Istanbul University – Cerrahpasa, Istanbul, Turkey, **3** Department of Internal Medicine, Cerrahpasa Medical Faculty, Istanbul University – Cerrahpasa, Istanbul, Turkey, **4** Department of Biostatistics and Medical Informatics, Halic University, Istanbul, Turkey, **5** Department of Microbiology, Cerrahpasa Medical Faculty, Istanbul University – Cerrahpasa, Istanbul, Turkey

\* nseyahi@yahoo.com

## Abstract

**Data Availability Statement:** All relevant data are within the manuscript and its Supporting Information files.

### Background

Recent data have suggested the presence of a reciprocal relationship between COVID-19 and kidney function. To date, most studies have focused on the effect of COVID-19 on kidney function, whereas data regarding kidney function on the COVID-19 prognosis is scarce. Therefore, in this study, we aimed to investigate the association between eGFR on admission and the mortality rate of COVID-19.

### Methods

We recruited 336 adult consecutive patients (male: 57.1%, mean age: 55.0±16.0 years) that were hospitalized with the diagnosis of COVID-19 in a tertiary care university hospital. Data were collected from the electronic health records of the hospital. On admission, eGFR was calculated using the CKD-EPI formula. Acute kidney injury was defined according to the KDIGO criteria. Binary logistic regression and Cox regression analyses were used to assess the relationship between eGFR on admission and in-hospital mortality of COVID-19.

### Results

Baseline eGFR was under 60 mL/min/1.73m$^2$ in 61 patients (18.2%). Acute kidney injury occurred in 29.2% of the patients. In-hospital mortality rate was calculated as 12.8%. Age-adjusted and multivariate logistic regression analysis (p: 0.005, odds ratio: 0.974, CI: 0.956–0.992) showed that baseline eGFR was independently associated with mortality. Additionally, age-adjusted Cox regression analysis revealed a higher mortality rate in patients with an eGFR under 60 mL/min/1.73m$^2$.

**Funding:** The authors received no specific funding for this work.

**Competing interests:** The authors have declared that no competing interests exist.

## Conclusions

On admission eGFR seems to be a prognostic marker for mortality in patients with COVID-19. We recommend that eGFR be measured in all patients on admission and used as an additional tool for risk stratification. Close follow-up should be warranted in patients with a reduced eGFR.

## Introduction

An outbreak of a novel coronavirus (severe acute respiratory syndrome coronavirus 2 [SARS-CoV-2]; also termed as COVID-19) has emerged from Wuhan city, China in December 2019 and spread to over 214 countries and territories worldwide within six months [1]. The first case of COVID-19 in Turkey was confirmed on March 10, 2020, and the World Health Organization declared the disease a pandemic on March 11. In severe cases, acute respiratory failure due to diffuse alveolar damage constitutes the main clinical characteristics of COVID-19, whereas kidneys are among the most common extrapulmonary targets of the virus [1–3].

Chronic kidney disease (CKD) is associated with alterations of innate and adaptive immunity [4]. When compared to the general population, both the pneumonia risk and the mortality rate due to pneumonia are increased in patients with CKD [5, 6]. In line with this information, the European Renal Association—European Dialysis and Transplant Association (ERA-EDTA) suggested that CKD patients possess an increased risk for COVID-19 and related mortality [7]. It has also been recently shown that acute kidney injury (AKI) was related with mortality in coronavirus infections, including COVID-19 [8]. However, formal studies that examine on admission kidney function on COVID-19 mortality is largely missing.

In this study, we primarily aimed to investigate the effects of kidney function on the prognosis of COVID-19, exclusively focusing on the estimated glomerular filtration rate (eGFR) on admission, and secondarily to determine the rate of AKI in COVID-19 patients.

## Materials and methods

The study protocol was approved by the Clinical Research Ethics Committee of Istanbul University Cerrahpasa (approval no: 2020–56318) and the Scientific Committee of the Ministry of Health (approval no: 2020-05-07T13_09_11). The study was conducted in accordance with the 1975 Declaration of Helsinki, as revised in 2013. The form of consent was not obtained because the data were analyzed anonymously. The ethics committee waived the requirement for informed consent.

Medical records of consecutive adult (>18 years) patients hospitalized between March 15 and May 1, 2020, were reviewed; outcomes data until June 1, 2020, were retrieved. The source of medical records was ISHOP (Istanbul University-Cerrahpasa Hospital Automation Program) electronic database system.

### Setting

The study was conducted in a tertiary care university hospital in Istanbul, where approximately 60% of all cases in Turkey were reported [9]. Istanbul University—Cerrahpasa, Cerrahpasa Medical Faculty hospital is one of the largest university hospitals in Istanbul, with a total of 897 hospital beds, 270 of which were allocated for the current pandemic.

## Patients

The diagnosis of COVID-19 was confirmed with at least one positive real-time reverse transcriptase-polymerase chain reaction (RT-PCR) test result in cases admitted with symptoms, signs and findings (laboratory / radiological) suggestive of COVID-19, according to the national guidelines [10].

Patients without any RT-PCR positivity, and those considered as 'possible' or 'probable' cases according to the Centers for Disease Control and Prevention (CDC) criteria were not included in the study (Fig 1) [11].

## Diagnostic methodology

Combined pharyngeal and nasopharyngeal swab samples were obtained for the RT-PCR assay. In cases who were followed with invasive mechanical ventilation in the intensive care unit (ICU), lower respiratory tract specimens were also obtained.

RNAs were extracted using a commercial kit (BioSpeedy Nucleic Acid extraction kit; Bioeksen R & D Technologies Ltd., Istanbul, Turkey), followed by the detection of COVID-19 RNA using a commercial RT-PCR kit (Bio-Speedy COVID-19 RT-qPCR kit; Bioeksen R & D Technologies Ltd., Istanbul, Turkey) that targets the RdRP gene of COVID-19 in the samples. Both kits were used according to the manufacturer's guidelines. The RT-PCR test was performed using 20 μl final volume using the following protocol: 5 min RT-PCR at 52˚C, 10sec initial denaturation step at 95˚C, followed by 40 cycles of 1 sec at 95˚C, and 30 sec at 60˚C. The Rotor-Gene Q 5plex HRM platform was used for amplification and detection.

## Data collection and design

Demographic, clinical and laboratory data were retrieved from the electronic database of the hospital by two different teams from the Nephrology and Infectious Diseases departments, cross-checked, and then were screened for duplicate records, missing and erroneous data. The National Public Health Data Management System database was used as an external data source, particularly to track the molecular test results. Data regarding medications were obtained from the medical records of the patients and via the National Health Insurance database when necessary.

Clinical data consisted of symptoms and vital findings including temperature (on admission) and oxygen saturation (lowest levels). Comorbidities were retrieved from medical records. Patients using antihypertensive drugs were accepted as hypertensive, while those using antidiabetic drugs were accepted as diabetic. Laboratory data consisted of measurements of serum urea, creatinine, uric acid, sodium, potassium, calcium, albumin, lactate dehydrogenase (LDH), liver function tests (AST, ALT), C-reactive protein (CRP), procalcitonin, ferritin, D-dimer, fibrinogen, creatine phospho-kinase (CPK), hematocrit (Htc), white blood cell (WBC), lymphocyte, platelet count (PLT) and urinalysis on admission. Additionally, discharge and peak creatinine values were also collected. Two patients were found to have been admitted more than once; for laboratory data, first admission values were recorded, while mortality data were obtained from their last admission.

The data underlying this article can be shared upon a reasonable request to the corresponding author.

## Study definitions

The date of hospital admission was accepted as the first day. The eGFR was calculated using the Chronic Kidney Disease Epidemiology Collaboration (CKD-EPI) formula [12].

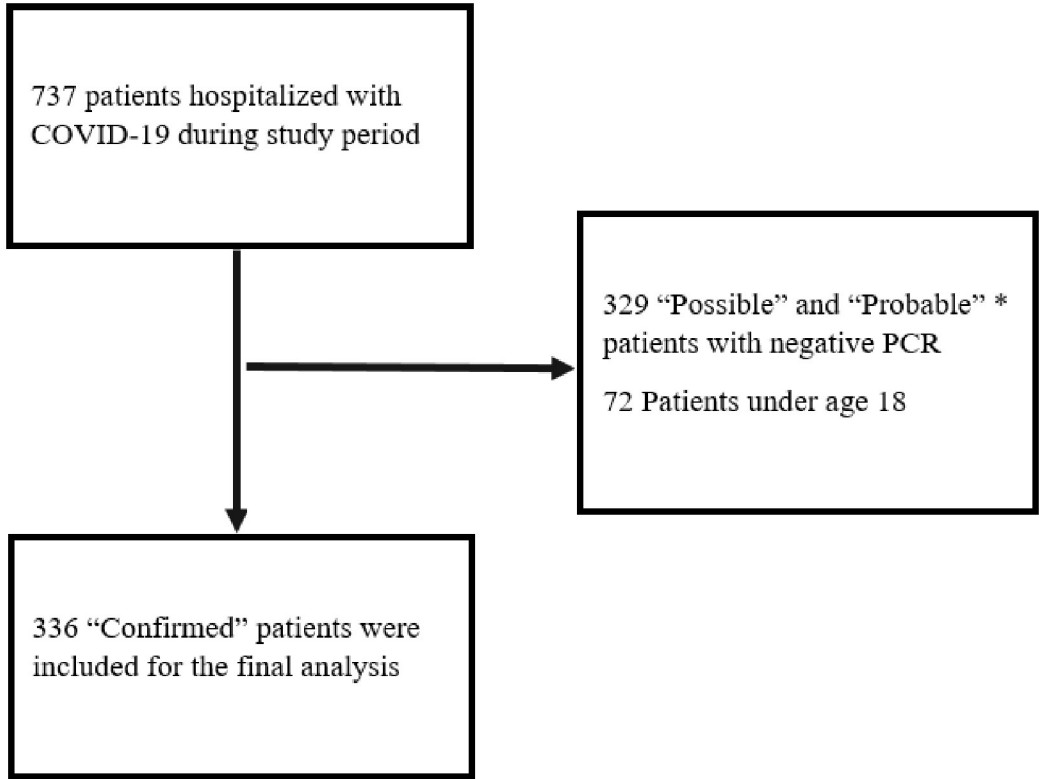

**Fig 1. Flow chart showing the selection of the patients.** *Definitions are based on the European Centre for Disease Prevention and Control [11].

Acute kidney injury was defined as an acute increase in the serum creatinine level of at least 0.3 mg/dl within 48 hours or a 50% increase in the serum creatinine level within 7 days from the baseline, according to the KDIGO guideline. Staging of the AKI was also performed according to the same guideline [13]. Complete renal recovery was defined as the regression of the discharge creatinine level to the baseline creatinine level and partial renal recovery was defined as a difference of less than 0.3 mg/dl between the baseline and discharge creatinine levels.

Body temperature was measured using non-contact infrared thermometers, and the presence of fever was defined as a temperature of more than 37.8˚C. Microscopic hematuria was accepted as the presence of ⩾1+ on dipstick urine testing or the presence of three or more erythrocytes per high-power field. Proteinuria was defined as the presence of ⩾1+ on dipstick urinalysis (at least 30–100 mg/dL).

## Statistical analysis

Data were expressed as mean±standard deviation, if not stated otherwise. Categorical variables were compared using the chi-square test and two-tailed exact significances (Fisher's exact test) were reported. Continuous variables were first analyzed for normality using the Kolmogorov-Smirnov test and then were compared using the paired samples t-test or the Mann-Whitney U test, when appropriate. Age-adjusted odds ratios were obtained with the use of logistic regression. Variables that were significantly associated with mortality in the age-adjusted analysis were used to construct a multivariate model. Binary logistic regression analysis with forward

conditional selection was used to evaluate the determinants of in-hospital mortality. Age-adjusted survival was calculated using the Cox regression analysis.

All tests were performed using SPSS for Windows, version 22.0 software (SPSS Inc., Chicago, IL, USA). P values of less than 0.05 were considered statistically significant.

## Results

### Baseline demographic, clinical and laboratory parameters

We examined a total of 336 unique patients. The demographic and clinical characteristics of the patients, stratified by baseline eGFR are shown in Table 1. Patients were usually middle to old age (median: 54; range: 18 to 94 years) and male gender (57.1%) was more prominent. The most common symptoms at admission werecough in 156 (46.4%), fever in 119 (35.4%), dyspnea in 89 (26.5%), weakness in 58 (17.3%), diarrhea in 29 (8.6%), nausea/vomiting in 26 (7.7%), myalgia in 26 (7.7%), headache in 14 (4.2%), expectorating in nine (2.7%), and smell and taste disorder in two (0.6%) patients. On admission systolic blood pressure was slightly higher for the eGFR <60mL/min/1.73 $m^2$ group, while there was no difference regarding on-admission diastolic blood pressure.

Hypertension was the most common comorbid disease, followed by diabetes mellitus (Table 1). Six patients were on chronic hemodialysis treatment, while only one patient had undergone kidney transplantation.

eGFR was <60 mL/min/1.73 $m^2$ in minority of the patients (18.1%). These patients were older, and comorbid conditions such as diabetes, hypertension and chronic obstructive pulmonary disease (COPD) were more common among them (Table 1). On-admission laboratory data of the patients are shown in Table 2. Besides differences among the laboratory parameters that were related to our kidney function-based classification, patients with an eGFR <60mL/min/1.73 $m^2$ had higher uric acid, potassium, LDH, ALT, CRP, procalcitonin, ferritin, D-dimer, WBC, and lower calcium, albumin, hematocrit levels, when compared to patients with an eGFR ≥60 mL/min/1.73 $m^2$. The lowest oxygen saturation was taken into consideration in all patients. The oxygen saturation level in patients with an eGFR <60 mL/min/1.73 $m^2$ was

**Table 1. Demographic and clinical characteristics of all patients and patients with a baseline eGFR <60 and ≥60 mL/min/1.73 $m^2$.**

| Characteristics | All patients (n = 336) | Patients with baseline eGFR <60 mL/min/1.73 $m^2$ (n = 61) | Patients with baseline eGFR ≥60 mL/min/1.73 $m^2$ (n = 275) | p |
|---|---|---|---|---|
| Age (years) | 55.0±16.0 | 69.9±12.9 | 51.7±14.7 | <0.001 |
| Gender, male | 192 (57.1) | 29 (47.5) | 163 (59.3) | 0.115 |
| Diabetes mellitus | 63 (18.8) | 23 (37.7) | 40 (14.5) | <0.001 |
| Hypertension | 120 (35.7) | 42 (68.9) | 78 (28.4) | <0.001 |
| Systolic blood pressure (mmHg)[1] | 122.4± 18.3 | 127.9±23.2 (n = 59) | 121.0±16.6 (n = 234) | 0.028 |
| Diastolic blood pressure (mmHg)[1] | 74.0± 10.8 | 75.3±11.5 (n = 59) | 73.7±10.6 (n = 234) | 0.254 |
| Bronchial asthma | 20 (6.0) | 1 (1.6) | 19 (6.9) | 0.142 |
| COPD | 19 (5.7) | 12 (19.7) | 7 (2.5) | <0.001 |
| Heart failure | 16 (4.8) | 6 (9.8) | 10 (3.6) | 0.050 |
| Malignancy | 31 (10.1) | 5 (9.8) | 26 (10.2) | 1.000 |

eGFR: estimated glomerular filtration rate, COPD: chronic obstructive pulmonary disease.

Data are expressed as mean±SD for quantitative parameters and n (%) for nominal parameters.

[1]Quantitative blood pressure data was available for 293 patients in the electronic health records. The remaining 43 were recorded as normal.

**Table 2. Laboratory findings of all patients and patients with a baseline eGFR <60 and ⩾60 mL/min/1.73 m².**

| Parameters | All patients (n = 336) | Patients with baseline eGFR <60 mL/min/1.73 m² (n = 61) | Patients with baseline eGFR ⩾60 mL/min/1.73 m² (n = 275) | p |
|---|---|---|---|---|
| Urea (mg/dL) | 37.6±31.0 | 80.6±50.7 | 28.0±10.4 | <0.001 |
| Creatinine (mg/dL) | 1.2±1.2 | 2.6±2.3 | 0.8±0.2 | <0.001 |
| Admission eGFR (mL/min/1.73 m²) | 83.1±28.7 | 34.7±17.6 | 93.8±17.5 | <0.001 |
| Peak creatinine (mg/dL) | 1.5±1.7 | 3.5±3.3 | 1.1±0.5 | <0.001 |
| Time to peak creatinine (day) | 4.1±4.6 | 4.3±5.2 | 4.1±4.4 | 0.691 |
| eGFR calculated using peak creatinine (mL/min/1.73 m²) | 70.6±30.7 | 26.8±16.2 | 80.3±24.0 | <0.001 |
| Discharge creatinine[1] (mg/dL) | 1.1±1.1 | 2.4±2.2 | 0.9±0.4 | <0.001 |
| Discharge eGFR[1] (mL/min/1.73 m²) | 83.5±31.0 | 38.7±23.6 | 93.5±22.4 | <0.001 |
| Uric acid[2] (mg/dL) | 5.1±2.0 | 7.4±2.7 | 4.6±1.4 | <0.001 |
| Sodium (mEq/L) | 137.7±3.8 | 137.0±4.9 | 137.8±3.6 | 0.229 |
| Potassium (mEq/L) | 4.3±0.5 | 4.5±0.8 | 4.3±0.5 | 0.029 |
| Calcium[1] (mg/dL) | 8.7±0.6 | 8.5±0.6 | 8.8±0.6 | <0.001 |
| Albumin[1] (g/dL) | 3.9±0.5 | 3.5±0.5 | 4.0±0.5 | <0.001 |
| LDH (U/L) | 298.6±287.4 | 361.8±280.1 | 284.6±287.6 | 0.016 |
| AST (U/L) | 38.6±73.7 | 67.3±163.6 | 32.2±23.5 | 0.614 |
| ALT (U/L) | 35.2±89.5 | 50.7±181.9 | 31.8±49.8 | 0.003 |
| CRP (mg/dL) | 57.9±72.6 | 97.5±88.1 | 49.1±65.7 | <0.001 |
| Procalcitonin[3] (ng/mL) | 1.4±9.3 | 2.7±8.8 | 1.1±9.4 | <0.001 |
| Ferritin[4] (ng/mL) | 506.8±889.6 | 892.0±1731.3 | 426.2±549.7 | 0.008 |
| D-dimer (ng/mL) | 1.6±3.1 | 3.2±4.3 | 1.2±2.6 | <0.001 |
| Fibrinogen[5] (mg/dL) | 443.8±165.1 | 458.6±160.3 | 440.5±166.3 | 0.467 |
| CPK[6] (U/L) | 157.5±359.1 | 128.6±125.4 | 164.0±392.7 | 0.950 |
| Hematocrit[1] (%) | 38.0±5.7 | 34.5±6.4 | 38.8±5.2 | <0.001 |
| White blood cell (/mm³) | 6602.7±3491.1 | 7845.9±4414.9 | 6326.9±3195.7 | <0.001 |
| Lymphocyte (/mm³) | 1519.0±1595.1 | 1226.2±706.6 | 1583.9±1725.7 | 0.113 |
| Platelet (/mm³) | 215087.1 ±150039.8 | 211426.2±115593.9 | 215899.1±156825.1 | 0.834 |

eGFR: estimated glomerular filtration rate, LDH: lactate dehydrogenase, AST: aspartate aminotransferase, ALT: alanine aminotransferase, CRP: C-reactive protein,

CPK: creatine phospho-kinase.

[1]There were missing data in less than three patients.

[2]Uric acid was measured in 325 patients.

[3]Procalcitonin was measured in 310 patients.

[4]Ferritin was measured in 318 patients.

[5]Fibrinogen was measured in 297 patients.

[6]CPK was measured in 301 patients.

Data are expressed as mean±SD.

lower when compared to those with eGFR >60 mL/min/1.73 m² (89.3±8.3% vs 92.4±5.9%, respectively; p:0.01). Urinalysis of the 67 patients confirmed that 23 patients had hematuria and 17 had proteinuria.

Drugs used for the management of COVID-19 are summarized in Table 3. Favipiravir was more commonly used in patients with an eGFR <60 mL/min/1.73 m². There were no statistically significant differences between the two groups (eGFR <60mL vs ≥60 mL/min/1.73 m²) regarding other drugs (Table 3).

**Table 3. Drugs used in the treatments of all patients and patients with a baseline eGFR<60 and ⩾60 mL/min/1.73 m².**

| Drug | All patients (n = 336) | Patients with baseline eGFR<60 mL/min/1.73 m² (n = 61) | Patients with baseline eGFR⩾60 mL/min/1.73 m² (n = 275) | p |
|---|---|---|---|---|
| Hydroxychloroquine | 332 (98.8) | 59 (96.7) | 273 (99.3) | 0.152 |
| Oseltamivir | 285 (84.8) | 48 (78.7) | 237 (86.2) | 0.166 |
| Azithromycin | 294 (87.5) | 51 (83.6) | 243 88.4) | 0.292 |
| Favipiravir | 169 (50.2) | 41 (67.2) | 128 (46.5) | 0.004 |
| Lopinavir/ritonavir | 27 (8.0) | 6 (9.8) | 21 (7.6) | 0.421 |
| Tocilizumab | 57 (17.8) | 9 (14.8) | 48 (17.5) | 0.708 |
| LWMH | 192 (57.3) | 40 (65.6) | 152 (55.3) | 0.115 |
| Glucocorticoid | 21 (6.3) | 2 (3.3) | 19 (6.9) | 0.390 |

eGFR: estimated glomerular filtration rate, LMWH: low-molecular-weight heparin.

Data are expressed as n (%).

## Prevalence of AKI, intensive care unit admission, and in-hospital mortality

Patients stayed in the hospital for 10 days on average. Acute kidney injury was detected in 98 patients (29.2%) and most (68.4%) of the AKI cases were Stage 1 (Table 4). Intensive care unit admission was necessary for 17.6% of the patients. During their hospitalization, 12.8% of the patients died.

Patients with an eGFR <60 mL/min/1.73 m² had longer hospital stays. Acute kidney injury was more common in patients with a baseline eGFR <60 mL/min/1.73 m². All three stages of AKI were also more common in patients with a baseline eGFR <60 mL/min/1.73 m² (Table 4). Continuous renal replacement therapy (RRT) was performed in four cases due to Stage 3 AKI, and in three of them, baseline eGFR was <60 mL/min/1.73 m². Thirty-four patients with AKI have died (34.7%). We observed complete renal recovery in 36 (36.7%) and partial renal recovery in 23 patients (23.5%). Discharge creatinine remained 0.3 mg/dL above admission creatinine in five patients (5.1%). In-hospital mortality was significantly lower (3.7%, p<0.001) in patients without AKI compared to that of patients with AKI.

The ICU admission and in-hospital death rates were significantly higher in patients with a baseline eGFR <60 mL/min/1.73 m² (Table 4). Specifically, three of the six patients who were on chronic hemodialysis treatment have also died, while one patient who had undergone kidney transplantation survived.

**Table 4. Incidence of acute kidney injury, intensive care unit admission and in-hospital mortality.**

| Characteristics | All patients (n = 336) | Patients with baseline eGFR <60 mL/min/1.73 m² (n = 61) | Patients with baseline eGFR ⩾60 mL/min/1.73 m² (n = 275) | p |
|---|---|---|---|---|
| Acute kidney injury | 98 (29.2) | 38 (62.3) | 60 (21.8) | <0.001 |
| Stage 1 | 67 (19.9) | 29 (47.5) | 38 (13.8) | <0.001 |
| Stage 2 | 16 (4.8) | 4 (6.6) | 12 (4.4) | |
| Stage 3 | 15 (4.5) | 5 (8.2) | 10 (3.6) | |
| Hospital stay (days) | 10.2±7.0 | 11.9±6.8 | 9.8±7.1 | 0.007 |
| ICU admission rate | 59 (17.6) | 21 (34.4) | 38 (13.8) | <0.001 |
| In-hospital mortality | 43 (12.8) | 21 (34.4) | 22 (8.0) | <0.001 |

eGFR: estimated glomerular filtration rate, ICU: intensive care unit.

Data are expressed as mean±SD for quantitative parameters and n (%) for nominal parameters.

## Determinants of in-hospital mortality

A total of 43 patients (12.8%) died during their hospital stay. Comparison of the demographic, clinical and laboratory findings of the deceased and living patients is given in Table 5. Patients

**Table 5. Demographic, clinical and laboratory findings of patients who died and those who survived.**

| Parameters | Patients who died (n = 43) | Those who survived (n = 293) | p | Age adjusted OR | CI 95% (min-max) | p |
|---|---|---|---|---|---|---|
| Age | 68.5±15.2 | 53.0±15.1 | <0.001 | NA | NA | NA |
| Gender, male | 27 (62.8) | 165 (56.3) | 0.510 | 1.543 | 0.756–3.149 | 0.233 |
| Diabetes mellitus | 15 (34.9) | 48 (16.4) | 0.006 | 0.554 | 0.262–1.172 | 0.123 |
| Hypertension | 22 (51.2) | 98 (33.4) | 0.027 | 1.288 | 0.605–2.745 | 0.512 |
| Systolic blood pressure (mmHg)[1] | 127.5± 25.9 (n = 41) | 121.6±16.7 (n = 252) | 0.190 | 1.006 | 0.988–1.025 | 0.509 |
| Diastolic blood pressure (mmHg)[1] | 73.3± 12.6 (n = 41) | 74.1± 10.5 (n = 252) | 0.535 | 0.994 | 0.963–1.027 | 0.725 |
| Bronchial asthma | 1 (2.3) | 19 (6.5) | 0.489 | 3.087 | 0.382–24.965 | 0.290 |
| COPD | 6 (14.0) | 13 (4.4) | 0.023 | 0.613 | 0.206–1.822 | 0.379 |
| Heart failure | 8 (18.6) | 8 (2.7) | <0.001 | 0.234 | 0.076–0.717 | 0.011 |
| Malignancy | 17 (39.5) | 14 (4.8) | <0.001 | 0.046 | 0.017–0.121 | <0.001 |
| Urea (mg/dL) | 71.4±58.3 | 32.6±20.5 | <0.001 | 1.021 | 1.011–1.031 | <0.001 |
| Creatinine (mg/dL) | 1.9±1.9 | 1.0±1.0 | <0.001 | 1.367 | 1.109–1.685 | 0.003 |
| Admission eGFR (mL/min/1.73m$^2$) | 59.7±37.4 | 86.5±25.6 | <0.001 | 0.983 | 0.970–0.996 | 0.010 |
| Peak creatinine (mg/dL) | 3.3±2.6 | 1.2±1.4 | <0.001 | 1.495 | 1.227–1.821 | <0.001 |
| Time to peak creatinine (day) | 7.6±7.2 | 3.6±3.8 | 0.001 | 1.140 | 1.068–1.217 | <0.001 |
| eGFR calculated using peak creatinine (mL/min/1.73 m$^2$) | 30.6±27.2 | 76.4±26.5 | <0.001 | 0.947 | 0.932–0.963 | <0.001 |
| Discharge creatinine (mg/dL) | 2.2±1.5 | 0.9±0.9 | <0.001 | 1.708. | 1.280–2.280 | <0.001 |
| Discharge eGFR (mL/min/1.73 m$^2$) | 44.9±37.0 | 89.2±25.5 | <0.001 | 0.960 | 0.948–0.973 | <0.001 |
| Uric acid (mg/dL) | 6.0±3.1 | 5.0±1.8 | 0.045 | 1.109 | 0.953–1.291 | 0.182 |
| Sodium (mEq/L) | 135.5±5.6 | 138.0±3.4 | 0.006 | 0.861 | 0.792–0.936 | <0.001 |
| Potassium (mEq/L) | 4.2±0.7 | 4.3±0.5 | 0.411 | 0.597 | 0.316–1.127 | 0.112 |
| Calcium (mg/dL) | 8.3±0.7 | 8.8±0.6 | <0.001 | 0.429 | 0.253–0.728 | 0.002 |
| Albumin (g/dL) | 3.3±0.5 | 4.0±0.5 | <0.001 | 0.075 | 0.032–0.175 | <0.001 |
| LDH (U/L) | 439.2±687.8 | 263.6±130.1 | <0.001 | 1.005 | 1.002–1.007 | <0.001 |
| AST (U/L) | 71.8±100.6 | 33.7±67.7 | <0.001 | 1.004 | 1.000–1.007 | 0.046 |
| ALT (U/L) | 66.8±215.4 | 30.5±48.4 | 0.863 | 1.005 | 0.999–1.011 | 0.102 |
| CRP (mg/dL) | 146.1±83.6 | 44.9±61.0 | <0.001 | 1.012 | 1.008–1.017 | <0.001 |
| Procalcitonin (ng/mL) | 3.1±9.3 | 1.1±9.3 | <0.001 | 1.008 | 0.980–1.037 | 0.569 |
| Ferritin (ng/mL) | 1192.0±1910.1 | 419.3±608.7 | <0.001 | 1.001 | 1.000–1.001 | 0.003 |
| D-dimer (ng/mL) | 5.0±6.6 | 1.1±1.7 | <0.001 | 1.279 | 1.126–1.453 | <0.001 |
| Fibrinogen (mg/dL) | 470.8±164.0 | 439.8±165.2 | 0.280 | 1.001 | 0.999–1.003 | 0.385 |
| CPK (U/L) | 182.1±162.4 | 153.7±381.1 | 0.033 | 1.001 | 1.000–1.001 | 0.192 |
| Hematocrit (%) | 32.3±6.3 | 38.9±5.1 | <0.001 | 0.833 | 0.779–0.891 | <0.001 |
| White blood cell (/mm$^3$) | 8576.7±5553.0 | 6313.0±2981.5 | 0.005 | 1.000 | 1.000–1.000 | 0.009 |
| Lymphocyte (/mm$^3$) | 883.7±532.7 | 1612.2±1676.3 | 0.005 | 0.998 | 0.997–0.999 | <0.001 |
| Platelet (/mm$^3$) | 176793.0±93824.5 | 220707.1±155927.0 | 0.073 | 1.000 | 1.000–1.000 | 0.054 |

NA: not applicable. COPD: chronic obstructive pulmonary disease, eGFR: estimated glomerular filtration rate, LDH: lactate dehydrogenase, AST: aspartate aminotransferase, ALT: alanine aminotransferase, CRP: C-reactive protein, CPK: creatine phospho-kinase.

[1]Quantitative blood pressure data was available for 293 patients in the electronic health records. The remaining 43 were recorded as normal.

Data are expressed as mean±SD for quantitative parameters and n (%) for nominal parameters. Please refer to foot-note of Table 2 for specific number of measurements of each laboratory parameter.

who died were older and commonly had comorbid conditions such as diabetes, hypertension, COPD, heart failure and malignancy. Laboratory parameters associated with kidney function were worse in the patients who died. Compared to the patients who survived, deceased patients had higher uric acid, LDH, AST, CRP, procalcitonin, ferritin, D-dimer, CPK, WBC, and lower sodium, calcium, albumin, hematocrit, and lymphocyte levels (Table 5). On-admission systolic and diastolic blood pressures were not different between patients who died or survived. The association between old age and COVID-19 mortality is well known, therefore, we calculated the age-adjusted odds ratios of the study parameters for mortality (Table 5). According to the age-adjusted analysis, heart failure, malignancy, kidney function parameters, and sodium, calcium, albumin, LDH, AST, CRP, ferritin, D-dimer, Hct, WBC and lymphocyte levels were associated with mortality.

We used Cox regression analysis to calculate the age-adjusted survival according to the eGFR group and prepared the survival curves accordingly (Fig 2). A baseline eGFR <60ml/min/1.73m$^2$ was associated with a reduced survival rate (p: 0.021, OR: 2.161, CI 95%, min-max: 1.121–4.167).

Finally, we constructed a multivariate model to determine the in-hospital mortality rate using the variables that were available on admission and were significantly associated with mortality in the age-adjusted analysis. We did not include the data that were not available on admission (peak creatinine, discharge creatinine, associated eGFRs and time to peak creatinine) and associates of eGFR (urea and creatinine) in our prediction model. According to the multivariate model, malignancy, eGFR, CRP and Hct levels on admission were independent determinants of mortality (Table 6).

## Discussion

We showed that eGFR on admission was an independent determinant of mortality in patients with COVID-19. The association of COVID-19 with kidney function can be addressed in two reciprocal ways, which are not mutually exclusive; first, the effect of kidney disease on the course of COVID-19 can be examined, and second, the effect of COVID-19 on kidney function and development of AKI can be examined. In this paper, we mainly examined the first part of this association. Additionally, we also examined the rate of AKI. To the best of our knowledge, most studies have focused on the second part of this association from a clinical or histopathological point of view [2, 14–19].

In addition to pulmonary infiltration, SARS-CoV-2 may have cytopathic effects in many organs, including renal tissue [3]. It has been reported that ACE2, the cell entry receptor of SARS-CoV-2, is expressed almost 100 times higher in the kidneys than in the lungs [20, 21]. The pathogenesis of kidney disease in patients with COVID-19 is probably multifactorial, including direct cytopathic effects on kidney tissue, endothelial damage, deposition of immune complexes and virus-induced cytokines or mediators [1, 15, 22]. Su et al. investigated postmortem findings of COVID-19 patients and found evidence of the direct cytopathic effect of COVID-19 on kidney tissue. Moreover, endothelial damage was a common finding in renal histopathological analyses of 26 COVID-19 patients, in the absence of interstitial inflammatory infiltrates [3]. COVID-19 infection could cause endothelial dysfunction and a hypercoagulation state. This condition is aggravated by hypoxia, which augments thrombosis by both increasing blood viscosity and hypoxia-inducible transcription factor-dependent signaling pathway [23]. Hirsch et al. reported the incidence of AKI in a large cohort consisting of 5,449 patients and suggested ischemic acute tubular necrosis as an important aetiology for AKI in COVID-19 [19].

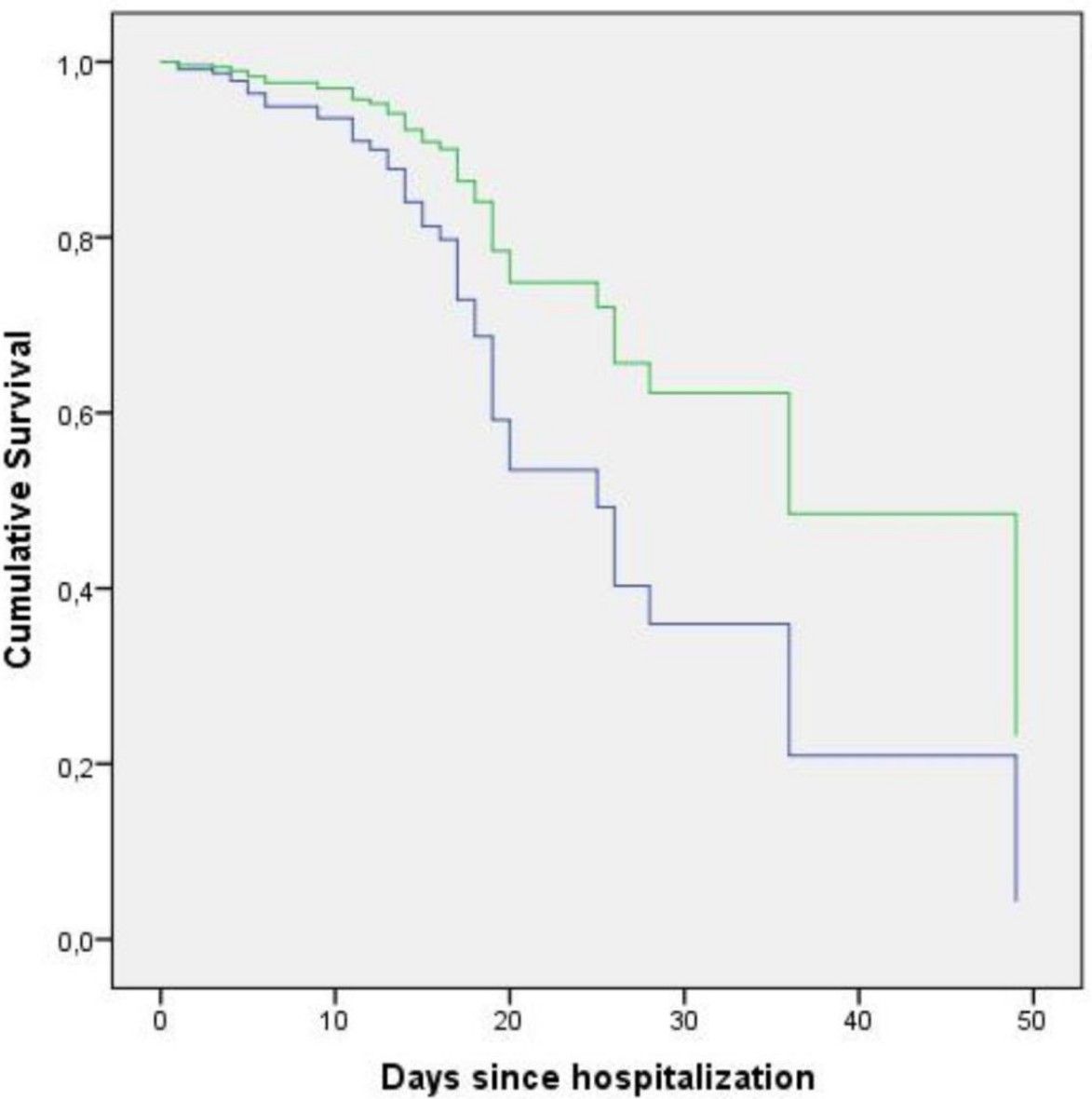

**Fig 2. Cumulative survival plots of the patients stratified by eGFR on admission.** The plots are prepared according to age-adjusted Cox regression analysis. The green line depicts patients with eGFR ⩾60 mL/min/1.73 m$^2$ and the blue line depicts patients with eGFR <60 mL/min/1.73 m$^2$.

**Table 6. Variables associated with mortality according to the multivariate binary logistic regression model.**

| Variables | Age-adjusted OR | CI 95% (min-max) | p |
|---|---|---|---|
| Malignancy | 29.412 | 7.194–125.000 | <0.001 |
| Admission eGFR (mL/min/1.73m$^2$) | 0.974 | 0.956–0.992 | 0.005 |
| CRP (mg/dL) | 1.012 | 1.005–1.018 | <0.001 |
| Hematocrit (%) | 0.879 | 0.796–0.972 | 0.012 |

eGFR: estimated glomerular filtration rate, CRP: C-reactive protein.

We report a high rate of AKI (29.2%) and a high mortality rate in patients with AKI (34.7%). According to previous clinical studies, the detection rate of AKI in patients with COVID-19 has been reported to vary between 0.5% and 36.6% [15, 17, 19]. Wang et al. claimed that COVID-19 was not associated with AKI [14]. Curiously, in that study, AKI was not reported even in the patients who died in the ICU. Cheng et al. reported AKI in 5.1% of a cohort of 701 patients [15]. The in-hospital death rate in their study was 16.1%, while it was calculated as high as 33.7% in those with elevated baseline serum creatinine levels. Another study from China examined 1,099 patients and found that mortality or ICU admission rates in patients with a higher creatinine level were higher (9.6%, n = 52) than those with normal creatinine levels (1%, n = 700) [17]. Chen at al. evaluated the characteristics of deceased COVID-19 patients and found that AKI was more frequent in patients who died (25%) when compared to those who survived (1%) [16]. In another study, Pei et al. reported that 6% of their patients experienced AKI [18], while Lim et al. found that the median age was higher in AKI patients [24]. Moreover, Lim et al. reported the highest rate of mortality in COVID-19 patients with AKI. The authors examined 164 hospitalized patients with COVID-19 and reported AKI in 18.3% of them, while in-hospital mortality was significantly higher in patients with AKI (56.7%). Hirsch et al. reported the highest AKI rate (36.6%) among COVID-19 patients and the mortality rate of this group was 35% [19]. Continuous veno-venous hemodialysis (CVVHD) has been suggested as a safe method for treatment COVID-19 patients with AKI who need RRT [25].

Studies that analyzed the relationship between AKI or peak creatinine and the prognosis of COVID-19 might be prone to look-ahead bias. Therefore, we want to emphasize the importance of analyzing the relationship between kidney function on admission and mortality. We exclusively focused on baseline eGFR, since data on the peak creatinine level or development of AKI during the hospital stay are not available at the time of admission and risk stratification based on those characteristics might lead to a look-ahead bias. Similar to our findings, Cheng et al. recently showed that the prevalence of kidney disease during hospitalization in patients with COVID-19 was high and was associated with in-hospital mortality [15]. However, their analysis was based on creatinine levels; they did not use eGFR as a prognostic marker.

Heterogeneity of the results regarding the development of AKI might be explained by differences in baseline demographics, comorbid conditions and respiratory disease severity [19]. Moreover, there were differences among countries regarding their response to the pandemic. Besides the differences in the extent and types of social isolation measures, treatment algorithms, modalities and drug use were also different. Additionally, there are demographic differences between countries; all these factors might also affect clinical end-points. Therefore, having data from different countries and geographic regions is important to better understand and manage COVID-19 globally.

The following points summarize Turkey's position regarding the baseline characteristics and its response to the pandemic. Turkey is one of the countries with a relatively young population (30.7% of the population is under the age of 20) and a wide health coverage through government programs [26, 27]. The number of hospital beds and ICUs per population is high [26]. Regarding COVID-19 treatment, early initiation of hydroxychloroquine was practiced, favipiravir use was widely adopted, and tocilizumab was used in all patients when indicated [10].

There are several limitations of our paper. First, urine analysis was not available in a large proportion of patients and we did not collect data on kidney imaging. Therefore, we might have overlooked some patients with CKD. Second, our follow-up duration was limited by the hospital stay period of the patients. The recovery patterns of kidney function might change during a longer follow-up. Third, we did not perform a formal power analysis to determine the

sample size. However, we recruited all eligible patients that were hospitalized. Fourth, our study was performed in a leading university hospital; it is possible that we might have recruited more severe patients. Finally, the generalizability of our results to other countries might be limited since countries have adopted different treatment guidelines according to local regulations and the availability of health resources.

In conclusion, eGFR on admission seems to be a prognostic marker for mortality in patients with COVID-19. We recommend that eGFR be measured in all patients on admission and used as an additional tool for risk stratification. Close follow-up might be warranted in patients with a reduced eGFR.

## Supporting information

**S1 File. COVID-19 dataset.** File containing dataset used for the main analysis.
(XLSX)

## Acknowledgments

We would like to thank secretary Gurbet Kaya for her valuable help during database editing.

## Author Contributions

**Conceptualization:** Sinan Trabulus, Rıdvan Karaali, Fehmi Tabak, Nurhan Seyahi.

**Data curation:** Sinan Trabulus, Cebrail Karaca, Mevlut Tamer Dincer, Ahmet Murt, Seyda Gul Ozcan, Rıdvan Karaali, Bilgul Mete, Alev Bakir, Mehmet Riza Altiparmak, Nurhan Seyahi.

**Formal analysis:** Sinan Trabulus, Cebrail Karaca, Mevlut Tamer Dincer, Ahmet Murt, Seyda Gul Ozcan, Alev Bakir, Nurhan Seyahi.

**Investigation:** Cebrail Karaca, Ilker Inanc Balkan, Mevlut Tamer Dincer, Ahmet Murt, Seyda Gul Ozcan, Bilgul Mete, Mert Ahmet Kuskucu, Mehmet Riza Altiparmak, Fehmi Tabak.

**Methodology:** Sinan Trabulus, Ilker Inanc Balkan, Rıdvan Karaali, Alev Bakir, Mert Ahmet Kuskucu, Mehmet Riza Altiparmak, Fehmi Tabak, Nurhan Seyahi.

**Supervision:** Sinan Trabulus, Ilker Inanc Balkan, Nurhan Seyahi.

**Writing – original draft:** Sinan Trabulus, Ilker Inanc Balkan, Mevlut Tamer Dincer, Ahmet Murt, Seyda Gul Ozcan, Nurhan Seyahi.

**Writing – review & editing:** Sinan Trabulus, Nurhan Seyahi.

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
