## [Decision Letter · Decision Letter 0]

31 Jul 2020

PONE-D-20-18434

Kidney function on admission predicts in-hospital mortality in COVID-19

PLOS ONE

Dear Dr. Seyahi,

Thank you for submitting your manuscript to PLOS ONE. After careful consideration, we feel that it has merit but does not fully meet PLOS ONE’s publication criteria as it currently stands. Therefore, we invite you to submit a revised version of the manuscript that addresses the points raised during the review process.

The introduction section is not easy to read. Available evidence should be more clearly summarized and the rationale should be more clearly stated. The manuscript needs an extensive English revision. 

We look forward to receiving your revised manuscript.

Kind regards,

Chiara Lazzeri

Academic Editor

PLOS ONE

Journal Requirements:

2. Thank you for including your ethics statement: 'The study protocol was approved by the local medical ethical committee (approval no: 2020-56318) and the Scientific Committee of the Ministry of Health (approval no: 2020-05-07T13_09_11).'

(a) Please amend your current ethics statement to include the full name of the ethics committee that approved your specific study. 

(b) Once you have amended this/these statement(s) in the Methods section of the manuscript, please add the same text to the “Ethics Statement” field of the submission form (via “Edit Submission”).

3. In the ethics statement in the manuscript and in the online submission form, please provide additional information about the patient records used in your retrospective study, including: a) whether all data were fully anonymized before you accessed them; b) the date range (month and year) during which patients' medical records were accessed; and c) the source of the medical records analyzed in this work (e.g. hospital, institution or medical center name). If the ethics committee waived the requirement for informed consent, please include this information.

4. We suggest you thoroughly copyedit your manuscript for language usage, spelling, and grammar. If you do not know anyone who can help you do this, you may wish to consider employing a professional scientific editing service.  

"NO"

6. Thank you for stating the following in your Competing Interests section: 

"NO"

7. Please upload a copy of Figures 1, 2, to which you refer in your text on pages 7 and 23. If the figures are no longer to be included as part of the submission please remove all reference to them within the text.

Reviewers' comments:

Reviewer's Responses to Questions

**Comments to the Author**

1. Is the manuscript technically sound, and do the data support the conclusions?

Reviewer #1: Yes

2. Has the statistical analysis been performed appropriately and rigorously? 

Reviewer #1: Yes

3. Have the authors made all data underlying the findings in their manuscript fully available?

Reviewer #1: Yes

4. Is the manuscript presented in an intelligible fashion and written in standard English?

Reviewer #1: No

5. Review Comments to the Author

Reviewer #1: Although this reviewer warmly welcomes this manuscript, some issues should be addressed:

-The rationale for the study is unclear as the introduction is a bit confusing, comprising several pieces of apparently unlinked information. A more integrated appraisal of the relevant literature would be appropriate to provide the context for the study.

-Figures were not available.

-The following reports should be mentioned:

J Clin Med. 2020 Jun 3;9(6):E1718; doi: 10.3390/jcm9061718; PMID: 32503180.

J Clin Med. 2020;9(5):1529. doi: 10.3390/jcm9051529. PMID: 32438617.

-Values of systolic and diastolic blood pressure should be included in the analysis.

-The key role of endothelial dysfunction in the pathophysiology of kidney injury in COVID-19 (J Clin Med. 2020 May 11;9(5):1417; doi: 10.3390/jcm9051417; PMID: 32403217) should be better discussed.

-English language (syntax, grammar, correct choice of words, correct use of adjectives and adverbs) should be substantially improved throughout the text. Professional assistance should be sought.

6. PLOS authors have the option to publish the peer review history of their article (what does this mean?). If published, this will include your full peer review and any attached files.

Reviewer #1: No

---

## [Author Response · Author response to Decision Letter 0]

17 Aug 2020

(This has also been sent as a separate file)

Editorial Review Board Comments:

1- Thank you for including your ethics statement: 'The study protocol was approved by the local medical ethical committee (approval no: 2020-56318) and the Scientific Committee of the Ministry of Health (approval no: 2020-05-07T13_09_11).'

(a) Please amend your current ethics statement to include the full name of the ethics committee that approved your specific study. 

(b) Once you have amended this/these statement(s) in the Methods section of the manuscript, please add the same text to the “Ethics Statement” field of the submission form (via “Edit Submission”).

Response: We made the suggested changes.

2. In the ethics statement in the manuscript and in the online submission form, please provide additional information about the patient records used in your retrospective study, including: a) whether all data were fully anonymized before you accessed them; b) the date range (month and year) during which patients' medical records were accessed; and c) the source of the medical records analyzed in this work (e.g. hospital, institution or medical center name). If the ethics committee waived the requirement for informed consent, please include this information.

Response: a) We anonymized the database during our analysis. b) We accessed the database between March 15 and May 1, 2020. c) The source of medical records was ISHOP (Istanbul University-Cerrahpasa Hospital Automation Program) electronic database system. d) The ethics committee waived the requirement for informed consent.

Response: The manuscript was edited by Edvard Tony Karakas from Daria Consulting, Turkey.

Response to Reviewer

Reviewer #1: Although this reviewer warmly welcomes this manuscript, some issues should be addressed:

1. The rationale for the study is unclear as the introduction is a bit confusing, comprising several pieces of apparently unlinked information. A more integrated appraisal of the relevant literature would be appropriate to provide the context for the study.

Response: We are sorry for this inconvenience. Apparently, a part of the introduction was deleted inadvertently during the submission. Following your comment, we became aware of this error and corrected the relevant section.

2. Figures were not available.

Response: We are sorry for this inconvenience. We believe that the figures were deleted because of a technical issue during the submission. We checked that they are correctly incorporated to the revised manuscript.

3. The following reports should be mentioned:

J Clin Med. 2020 Jun 3;9(6):E1718; doi: 10.3390/jcm9061718; PMID: 32503180.

J Clin Med. 2020;9(5):1529. doi: 10.3390/jcm9051529. PMID: 32438617.

Response: Thank you for providing us those important studies. We cited them in the revised manuscript.

4. Values of systolic and diastolic blood pressure should be included in the analysis.

Response: We collected the systolic and diastolic blood pressure records from our hospital electronic database and we made the necessary analysis in the revised manuscript.

5. The key role of endothelial dysfunction in the pathophysiology of kidney injury in COVID-19 (J Clin Med. 2020 May 11;9(5):1417; doi: 10.3390/jcm9051417; PMID: 32403217) should be better discussed.

Response: Thank you for providing us this important paper. We extended the Discussion section with the help of this paper.

6. English language (syntax, grammar, correct choice of words, correct use of adjectives and adverbs) should be substantially improved throughout the text. Professional assistance should be sought.

Response: With the assistance of a professional service, we revised the manuscript regarding language issues.

---

## [Editor Report · Decision Letter 1]

24 Aug 2020

Kidney function on admission predicts in-hospital mortality in COVID-19

PONE-D-20-18434R1

Dear Dr. Seyahi,

We’re pleased to inform you that your manuscript has been judged scientifically suitable for publication and will be formally accepted for publication once it meets all outstanding technical requirements.

Kind regards,

Chiara Lazzeri

Academic Editor

PLOS ONE
---

## [Editor Report · Acceptance letter]

27 Aug 2020

PONE-D-20-18434R1 

Kidney function on admission predicts in-hospital mortality in COVID-19 

Dear Dr. Seyahi:

I'm pleased to inform you that your manuscript has been deemed suitable for publication in PLOS ONE. Congratulations! Your manuscript is now with our production department. 

Kind regards, 

on behalf of

Dr. Chiara Lazzeri 

Academic Editor

PLOS ONE